# Cyclophobic Reinforcement Learning

## Abstract

In environments with sparse rewards, finding a good inductive bias for exploration is crucial to the agent's success. However, there are two competing goals: novelty search and systematic exploration. While existing approaches such as curiosity-driven exploration find novelty, they sometimes do not systematically explore the whole state space, akin to depth-first-search vs breadth-first-search. In this paper, we propose a new intrinsic reward that is cyclophobic, i.e. it does not reward novelty, but punishes redundancy by avoiding cycles. Augmenting the cyclophobic intrinsic reward with a sequence of hierarchical representations based on the agent's cropped observations we are able to achieve excellent results in the MiniGrid and MiniHack environments. Both are particularly hard, as they require complex interactions with different objects in order to be solved. Detailed comparisons with previous approaches and thorough ablation studies show that our newly proposed cyclophobic reinforcement learning is vastly more efficient than other state of the art methods.

## 1 Introduction

Exploration is one of reinforcement learning's most important problems. Learning success largely depends on whether an agent is able to explore its environment efficiently. Random exploration is explores all possibilities but at great costs, since it possibly revisits states very often. More efficient approaches use intrinsic rewards based on curiosity to enforce focusing on novelty, which often leads to great results, but at the price of possibly not exploring all corners of the environment systematically. Ideally, we would pursue both goals: novelty search and systematical exploration.

How can we favor novelty while ensuring that the whole environment is systematically explored? To achieve this, we propose *cyclophobic reinforcement learning* which is based on the simple idea of avoiding cycles during exploration. More precisely, we define a negative intrinsic reward that penalizes redundancy in the exploration history. This idea is further enhanced by applying it to several hierarchical views of the environment. The notion of redundancy can be defined relative to cropped views of the agent: while cycles in the global view induces cycles in the corresponding narrow view, the converse is not the case. E.g., a MiniGrid agent turning four times to the left produces a cycle in state space that we would like to avoid everywhere. This cycle is visible in the global view, but penalizing it does not avoid it in other locations. However, with a hierarchy of views, we record a cycle also in some smaller view, which allows us to transfer this knowledge to any location in the global view and hereby to avoid never experienced cycles. Similarly, an interesting object such as a key, produces less cycles in a smaller view than some other object (since the key can be interacted with). Thus the probability of picking up the key increases, since other *less interesting* observations produce more cycles (e.g a wall). Thus, we are defining cycles relative to a hierarchy of view to get a transferable definition of redundancy.

**Contributions:**

1. We introduce cyclophobic reinforcement learning as a new paradigm for efficient exploration in hard environments (e.g. with sparse rewards). It is based on a new cyclophobic intrinsic reward for systematic exploration applied to a hierarchy of views. Instead of rewarding novelty, we are avoiding redundancy by penalizing cycles, i.e. repeated state/action pairs in the exploration history. Our approach can be applied to any MDP for which a hierarchy of views can be defined.

2. In the sparse-reward settings of the MiniGrid and MiniHack environments we thoroughly evaluate cyclophobic reinforcement learning and can show that it achieves excellent results compared to existing methods, both for tabula-rasa and transfer learning. Preliminary results show cyclophobia works for both tabular and neural agents.

3. In an ablation study we provide deeper insights into the interplay of the cyclophobic intrinsic reward and the hierarchical state representations.

**Notation:** We define an MDP as a tuple $(\mathcal{S}, \mathcal{A}, \mathcal{P}, \mathcal{R}, \gamma)$, where the agent and the environment interact continuously at discrete time steps $t = 0, 1, 2, 3, \ldots$ We define the state an agent receives from the environment as a random variable $S_t \in \mathcal{S}$, where $S_t = s$ is some representation of the state from the set of states $\mathcal{S}$ at timestep $t$. From that state, we define a random variable for the agent selecting an action $A_t \in \mathcal{A}$ where $A_t = a$ is some action in the possible set of actions $\mathcal{A}$ for the agent at timestep $t$. This action is selected according to a policy $\pi(a \mid s)$ or $\pi(s)$ if the policy is deterministic. One time step later as a consequence of its action, the agent receives a numerical reward which is a random variable $R_{t+1} \in \mathbb{R}$, where $R_{t+1} = r$ is some numerical reward at timestep $t + 1$. Finally, the agent finds itself in a new state $S_{t+1}$. Furthermore we define a POMDP $(\mathcal{S}, \mathcal{A}, \mathcal{O}, \mathcal{P}, \mathcal{R}, \mathcal{Z}, \gamma)$ as a generalization of an MDP in the case the true state space $\mathcal{S}$ is unknown. That is, the agent sees the state $s \in \mathcal{S}$ through an observation $o \in \mathcal{O}$, where an observation function $\mathcal{Z} : \mathcal{S} \to \mathcal{O}$ maps the true state to the agent's observation.

## 2 Building Blocks

We begin by first defining the cyclophobic intrinsic reward and hierarchical state representations as they are the building blocks of our method. Finally, we define a policy which combines the intrinsic and extrinsic rewards together with the hierarchical state representations to form a global policy which the agent acts upon.

### 2.1 Cyclophobic Intrinsic Reward

For efficient exploration, redundancy must be avoided. A sign for redundancy is when states are repeatedly explored, with other words, when the agent encounters cycles in the state space instead of focusing on novel areas. To guide the exploration, we will penalize cycles using a cycle penalty, which we call cyclophobic intrinsic reward (a negative intrinsic reward). This avoids redundancy such that uninteresting parts of the state-action space are discarded quickly. For instance, if an agents gets stuck in some area, typically, it is facing numerous cycles. In such a situation we would like the agent to assign penalties to the repeating state-action pairs in order to focus on more promising parts of the state space that do not cause immediate repetition.

Formally, let us assume that we have per episode a history of previous state-action pairs $\mathcal{H}_{\text{episodic}} = \{(s_1, a_1), (s_2, a_2), \ldots, (s_t, a_t)\}$ and we are currently at the state-action pair $(s_{t+1}, a_{t+1})$. We say that we have encountered a cycle if the current state-action pair appeared already in the history, i.e. $(s_{t+1}, a_{t+1}) \in \mathcal{H}_{\text{episodic}}$. For its first repeated occurrence, we will penalize the state-action pair $(s_t, a_t)$ (just before the cycle) by negative one,

$$r_{\text{cycle}}(s, a) = -1. \tag{1}$$

If a cycle is encountered multiple times, e.g. $l$ times, the cycle penalty is $-l$. That is, during exploration the propagated cycle penalty can penalize indefinitely. For pairs $(s, a)$ that have not created a cycle, $r_{\text{cycle}}(s, a) = 0$.

**Learning the cycle penalties.** In principle, the cycle penalty can be combined with any reinforcement learning algorithm. However, we explain how it can be built into the SARSA update rule, since the latter will propagate the penalty across the trajectory. To learn the Q-function, we are employing the standard SARSA update rule,

$$Q(s, a) \leftarrow (1 - \eta)\, Q(s, a) + \eta \big[ r(s, a) + \gamma\, Q(s', a') \big] \tag{2}$$

with $(s, a, r_{\text{ex}}, s', a')$ being a transition, $\eta$ being step size and where the total reward $r(s, a)$ for the state-action pair $(s, a)$ is the weighted sum of the extrinsic reward from the environment $r_{\text{ex}}$ and the cyclophobic intrinsic reward $r_{\text{cycle}}(s, a)$ defined above,

$$r(s, a) = r_{\text{ex}} + \rho\, r_{\text{cycle}}(s, a). \tag{3}$$

where $\rho$ trades off extrinsic and intrinsic rewards.

### 2.2 Hierarchical State Representations

Besides penalizing cycles, the second key idea of this paper is to consider a hierarchy of state representations. For this, we repeatedly crop the agent's observations to induce additional partially

observable Markov decision processes (POMDPs). In general, restricting the view leads to ignoring information about the environment. Surprisingly, in combination with the cyclophobic intrinsic reward, we gain additional information about the structure of the environment as we get different Q-functions for different POMDP's. The relevant insight is that on limited views lower down the hierarchy, trajectories can contain cycles that have not been experienced on views higher up the hierarchy. These cycles in smaller views represent transferable knowledge about the structure of the environment. E.g. in the MiniGrid environment (see Section B.1) encountering a wall in the smallest view will cause a cycle, capturing that running into a wall is counterproductive. On larger views this knowledge is only available directly, if we tried all walls everywhere. So, the the smaller views capture relevant invariances that also apply to the larger views. In general MDPs it might not be obvious how to define hierarchical state representations. Ideally, the agent has some sort of "location" to allow "cropped" views. Additionally, local properties of the state space that result in cycles are transferable to the whole state space, so that learning to avoid cycles helps for efficient exploration. For grid world-like environments, the hierarchical state representations are easily definable, since the agent has a defined location that can be used to define smaller neighborhoods typically corresponding to limited views of the agent.

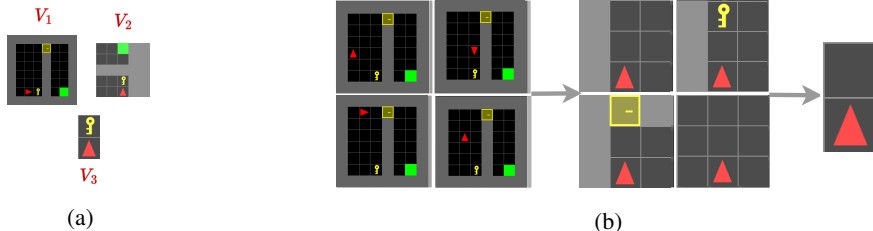

(a)  (b)

Figure 1: **Hierarchical views allow us to transfer relevant invariances about the environment.** (a) The three different representations are obtained by cropping the observation for each state-action pair $(s, a)$. $V_1$ is the full view where the agent sees the whole environment or the largest portion of it. $V_2$ is an intermediate cropped representation of the agent's view that helps the agent generalize by reusing familiar observations . $V_3$ is the most restricted view where the agent only sees what is immediately in front of it. (b) Through the views $V_1$ to $V_k$ the amount of cycles continuously increases as the observations in the higher views can be mapped multiple times to the same observation in the lower view. This naturally leads to each view being a separate POMDP describing the true MDP.

To discuss the roles of the different views more precisely, let's consider three such cropped views which consist of the global view $V_1$, the intermediate view $V_2$ and the smallest possible detail view $V_3$ (see Figure 1a). Each view gives a new, typically more limited, perspective of the true state. As shown in Figure 1b, each view induces a different set of cycles, for instance a sequence of actions which do not lead to a cycle in the full view $V_1$, might lead to a cycle in a smaller view $V_3$, because in the latter, more states are mapped to the same observation. Thus, the different views provide different types of information which are useful for learning and allow the agent to focus on different properties of the environment.

In general, we can have an arbitrary number of views. Each view $V_i$ induces a POMDP

$$\mathcal{V}_i = (\mathcal{S}, \mathcal{A}, \mathcal{O}^{V_i}, \mathcal{P}, \mathcal{R}, \mathcal{Z}^{V_i}, \gamma) \tag{4}$$

each having their own set of observations $\mathcal{O}^{V_i}$ and observation function $\mathcal{Z}^{V_i} : \mathcal{S} \to \mathcal{O}^{V_i}$. All POMDP's $\mathcal{V}_1, \mathcal{V}_2, \ldots$ operate on the same state space $\mathcal{S}$, however they have different sets of observations $\mathcal{O}$ and corresponding observation functions. General POMDPs can have a probabilistic observation function. In our case, the observations are deterministic functions of the full view, e.g. the observations for the $i$th POMDP of state $s$ is

$$o^{V_i} = \mathcal{Z}^{V_i}(s) \tag{5}$$

which corresponds to the cropping for the view $V_i$. Hereby we create partial representations of the state space that allow us to identify invariances in the environment by looking at the same true state $s$ through different perspectives.

Note that, POMDPs are normally used to model uncertain observations. That is, they are a generalization of MDPs where the true state is not observable. Here, the POMDP idea is used to model different

views of a fully observable state space. Therefore, we do not seek to solve a POMDP problem, but rather we extend a regular MDP to get redundant hierarchical representations.

## 2.3 A CYCLOPHOBIC POLICY FOR HIERARCHICAL STATE REPRESENTATIONS

In the following, we describe how the cyclophobic intrinsic reward and the hierarchical state representations can be combined into a policy that exploits the cyclophobic inductive bias. For this, we define several Q-functions along the views, and combine them as a weighted sum. The weights are determined by counts of the observations in each observation set $\mathcal{O}^{V_i}$ which we explain next.

**Mixing coefficients.** Many strategies for defining mixing coefficients are possible. For concreteness, we follow in this paper a simple schema, where we determine the weights from the observation counts, which are obtained from the history of states visited throughout training,

$$\mathcal{H}_{\text{all}} = \{s_1, s_2, \ldots, s_T\}. \tag{6}$$

Note that $\mathcal{H}_{\text{all}}$ contains all states that have been visited in the training so far, which is different from the states in the episodic history $\mathcal{H}_{\text{episodic}}$ that was used in Sec. 2.1 to define cycles. Denoting the corresponding views of the history as $o_t^{V_i} = \mathcal{Z}^{V_i}(s_t)$, the counts for view $V_i$ are

$$N(o_1^{V_i}), N(o_2^{V_i}), \ldots, N(o_T^{V_i}). \tag{7}$$

where $N$ counts the number of times the observation $o_t^{V_i}$ has been encountered. The raw counts are normalized by their maximum (maximum for simplicity, other normalizations are possible), because the smaller views have higher counts than the bigger views. The weights should be large for states that have been seen less often, so we subtract the normalized counts from one,

$$\alpha_*^{V_i}(o_t^{V_i}) = 1 - \frac{N(o_t^{V_i})}{\max(N(o_1^{V_i}), N(o_2^{V_i}), \ldots, N(o_T^{V_i}))}. \tag{8}$$

The previous formula allows us to compute the weights for each respective view $V_i$ for a single state $s_t$,

$$\alpha_*(s_t) = [\alpha_*^{V_1}(\mathcal{Z}^{V_1}(s_t)), \ldots, \alpha_*^{V_k}(\mathcal{Z}^{V_k}(s_t))] = [\alpha_*^{V_1}(o_t^{V_1}), \ldots, \alpha_*^{V_k}(o_t^{V_k})] \tag{9}$$

where $\alpha_*(s_t)$ is a vector. Finally, the softmax operator turns these weights into a vector of probabilities,

$$\alpha(s_t) = \text{softmax}(\alpha_*(s_t)) = [\alpha_1(s_t), \ldots, \alpha_k(s_t)]. \tag{10}$$

The entries of this vector are denoted by $\alpha_i(s_t)$ and are used to define the cyclophobic Q-function in the next section. The definition of $\alpha$ can be extended to all state $s$ by setting it to zero for unseen states, i.e. $\alpha(s) = 0$ (zero-vector) for $s \notin \mathcal{H}_{\text{all}}$.

In general, larger views in the hierarchy will have bigger entries in $\alpha(s_t)$ as the observations repeat less often than in the smaller views. Thus $\alpha(s_t)$ gives more weight to the larger views than the smaller ones. However, this is compensated by the cyclophobic intrinsic reward since it is triggered far more in the smaller views than the larger views.

**Cyclophobic Q-function.** To combine all views into a single global policy we define a mixture over the different Q-functions learned with the cyclophobic intrinsic reward as defined in Section 2.1. For view $V_i$, we define Q-function as

$$Q(o^{V_i}, a) \leftarrow (1 - \eta)\, Q(o^{V_i}, a) + \eta\big[r(o^{V_i}, a) + \gamma Q(o'^{V_i}, a')\big]. \tag{11}$$

This follows from our argumentation in Section 2.2, where we replace the state $s$ in Equation 2 by the observations $o^{V_i}$ in their respective views. Then we can define cyclophobic Q-function as the mixture of the Q-functions of each view,

$$Q_{\text{cycle}}(s, a) = \sum_i \alpha_i(s)\, Q(\mathcal{Z}^{V_i}(s), a) = \sum_i \alpha_i(s)\, Q(o^{V_i}, a). \tag{12}$$

Note that the mixing coefficients $\alpha_i(s_t)$ are only non-zero for states $s_t$ that appeared in the global history $\mathcal{H}_{\text{all}}$. Thus the cyclophobic Q-function is zero for states $s \notin \mathcal{H}_{\text{all}}$ not encountered.

**Cyclophobic policy.** Finally, the cyclophobic policy is defined to the greedy action for the cyclophobic Q-function, i.e.

$$\pi(s) = \arg\max_a Q_{\text{cycle}}(s, a) = \arg\max_a \sum_i \alpha_i(s) Q(o^{V_i}, a). \tag{13}$$

Having normalized the counts within each view ensures comparability of the counts. This ensures that Q-values from rare observations i.e. more salient have a larger effect on deciding the action for the policy $\pi$. In an ablation study in Section 3 we show that the combination of the cyclophobic intrinsic reward and hierarchical state representations is crucial to the methods success.

## 3 EXPERIMENTS

Our experiments are inspired by Parisi et al. (2021) and Samvelyan et al. (2021). We test in environments where the causal structure is complex and the binding problem (Greff et al., 2020), (van Steenkiste et al., 2019) arises. That is, where some form of disentangled representation of the environments plays an important role for efficiently finding solutions.

**Environments:** We test our method on the MiniGrid and MiniHack environments:
- The MiniGrid environment (Chevalier-Boisvert et al., 2018) consists of a series of procedurally generated environments where the agent has to interact with several objects to reach a specific goal. The MiniGrid environments pose currently a benchmark for the sparse reward problem, since a reward is only given when reaching the final goal state.
- The MiniHack environment (Samvelyan et al., 2021) is a graphical version of the NetHack environment (Küttler et al., 2020). We select environments from the *Navigation* and *Skill* tasks. The MiniHack environment has a richer observation space by containing more symbols than the MiniGrid environments and a richer action space with up to 75 different actions. While not necessarily tailored to the sparse reward problem as the MiniGrid environment, the high state-action complexity makes it one of the most difficult environments for exploration.

**State encoding:** For both environments we choose five croppings of the original full view. The views $V_1, V_2, V_3, V_4, V_5$ are of grid size $9 \times 9, 7 \times 7, 5 \times 5, 3 \times 3$ and $2 \times 1$. In principle we could also include the full view. However, in the experiments the performance was much better when we limit ourselves to the partial views. Furthermore, every part of the grid that is behind of the wall from the agent's perspective is occluded (note that this is not the case in Figure 1a for visualization purposes). Intuitively, limiting the views allows the agent to ignore irrelevant details that are far away.

Next, the views are mapped to hashcodes (we use the open source *xxhash* library). That is, we have a hash function $g$ that maps observation $o_t^{V_i}$ to a hashcode $h_t^{V_i} = g(o_t^{V_i})$. This helps us to quickly check for cycles as we only need to check whether two hashcodes are equal.

For the MiniHack environment, a text prompt is an integral part of the current state. So, for MiniHack, the hashcode for a state is the concatenation of the hashcodes of the observation $o_t^{V_i}$ and the text prompt $m_t$, where $m_t \in \mathbb{R}^k$ is an encoding of a string of length $k$, i.e.

$$h_t = g(o_t) + g(m_t). \qquad (\text{"+" denoting concatenation}) \tag{14}$$

**Training setup and baselines:** We train each agent for three runs using different seeds for every run. For transfer learning we use "DoorKey" and MultiEnv("MultiRoom-N4-S5", "KeyCorridorS3R3", "BlockedUnlockPickup") for pretraining. For transfer learning, during pretraining we save the extrinsic rewards in a second separate Q-table in addition to the main Q-table which contains values of equation 2. These extrinsic rewards in the second Q-table are then used at the beginning of transfer learning for each environment, while we continue to use the cyclophobic intrinsic reward when doing transfer learning.The baselines for the MiniGrid experiments are provided by Parisi et al. (2021) and allow us to compare our method to C-BET (Parisi et al., 2021), Random Network Distillation (Ostrovski et al., 2017), RIDE (Raileanu & Rocktäschel, 2020) and ICM (Pathak et al., 2017). For the MiniHack environment we compare our results to the baselines presented by Samvelyan et al. (2021) which include IMPALA (Espeholt et al., 2018), RIDE (Raileanu & Rocktäschel, 2020) and Random Network Distillation (Burda et al., 2018). For the skill tasks the only available baseline is IMPALA (Espeholt et al., 2018).

**Evaluation metric:** While during learning the intrinsic reward based on cyclophobia plays the essential role, the ultimate goal is to maximize the extrinsic reward that is provided by the environment. Thus for comparison, we have to plot the extrinsic reward the agent receives for each episode. The reward ranges from 0 to 1.

## 3.1 MINIGRID

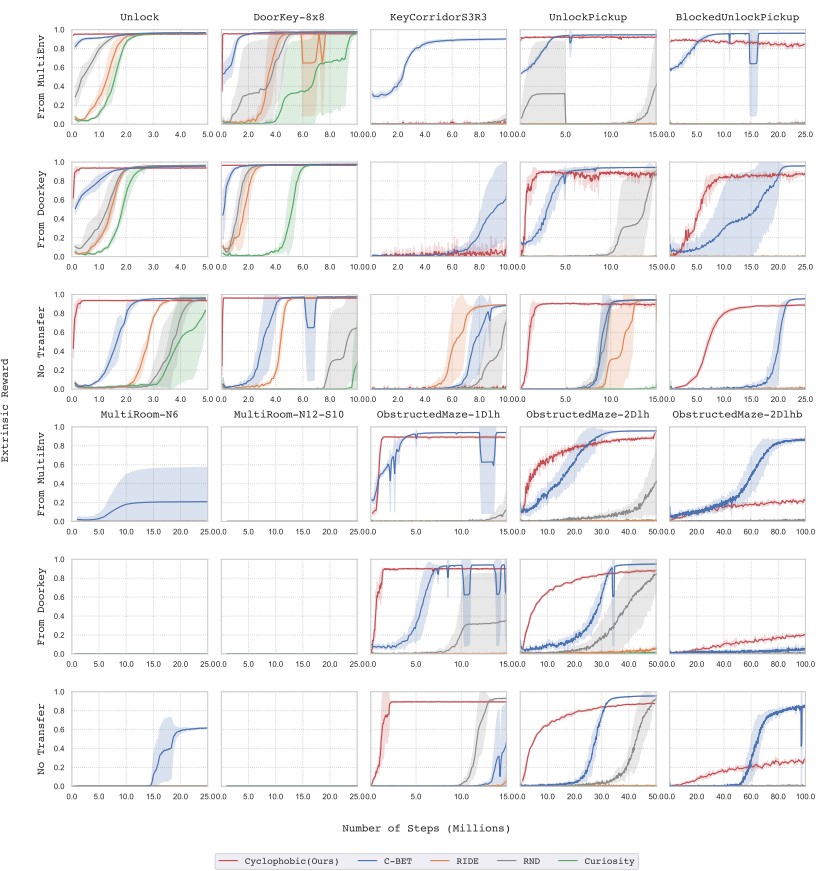

Figure 2: **MiniGrid:** We converge faster than C-BET (Parisi et al., 2021) in many MiniGrid environments with and w/o pretraining. The hierarchical state representations and cyclophobic intrinsic reward is extremely quick to converge which shows efficient exploration and the usefulness of the cropped representations. Furthermore we are also able to transfer knowledge by pretraining on other environments.

To test our method in the MiniGrid environment we choose the same training setup of Parisi et al. (2021). That is, we determine the performance of the agent with no pretraining and when pretraining from other environments as explained in the previous section. Figure 2 shows the agent's performance when training from scratch (rows three and six) and when transferring knowledge from pretrained environments (bottom three rows).

- **Learning from scratch** (rows three and six): In three out of six environments, our proposed method converges much faster than the competitors, including C-BET (Parisi et al., 2021). Note that for some environments our x-axis is shorter than in Parisi et al. (2021), since our method converges much faster. Only "KeyCorridorS3R3", "MultiRoom" and "ObstructedMaze-2Dlhb" pose significant challenges to our approach, because our method is tabular and thus cannot deal with too many object variations in the environment (e.g. random color changes). Furthermore, the "MultiRoom" environment proves challenging for all environments with only C-BET managing to reach convergence, while we are able to fetch some rewards. This is due to the large amount of observations the different corridors produce. In the Appendix A.2 our approach also excels in the "KeyCorridorS3R3", "ObstructedMaze-2Dlhb" and more difficult environments once we remove the colors, e.g. "KeyCorridorS4R3", "KeyCorridorS5R3", "KeyCorridorS6R3", "ObstructedMaze-1Q".
- **Transferring knowledge** (rows one, two, four and five): Having trained on one environment can we transfer knowledge to a different one? In some environments the transfer even im-

proved the results from the "no transfer" setup (see "Unlock", "Doorkey", "UnlockPickup", "BlockedUnlockPickup", "ObstructedMaze-1Dlh") and never deteriorated performance.

## 3.2 MINIHACK

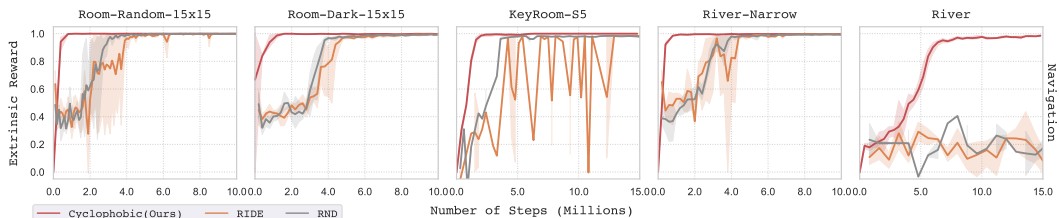

Figure 3: **MiniHack Navigation:** The agent converges quicker in the *Navigation* task than the intrinsic curiosity baselines such as RIDE (Raileanu & Rocktäschel, 2020) and Random Network Distillation (Burda et al., 2018). This corroborates our hypothesis, that avoiding cycles is essential for quick exploration.

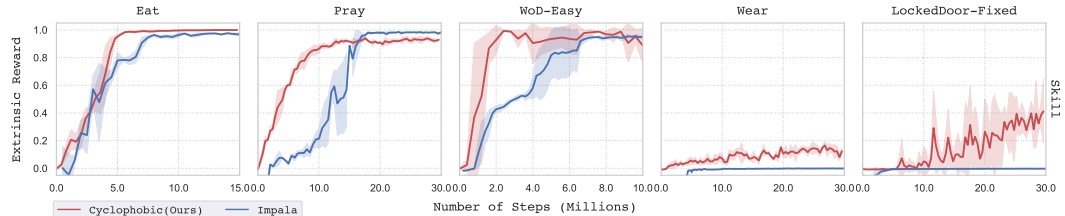

Figure 4: **MiniHack Skill:** We converge quicker than IMPALA (Espeholt et al., 2018) in the *Skill* task. The *Skill* task defines over 75 different actions the agent must learn to use making it one of the most difficult sets of environments of the MiniHack suite.

To push our approach to its limits, we also tackle some of the MiniHack environments which requires the agent to learn a large skill set.

- **Navigation task:** For the simpler navigation tasks shown in Figure 3, our method converges quicker than intrinsic curiosity driven methods such as RIDE and RND. Especially, in the "River" environment, only our cyclophobic agent is able to converge and solve the environment. However, of course there are environments such as "RoomUltimate" where our approach fails due to its tabular style, which limits the complexity of the environment.
- **Skill task:** For the *Skill* task the only available baseline is IMPALA which is not based on intrinsic curiosity. Here we are also vastly superior, even collecting extrinsic rewards when the baseline can not ("Wear" and "LockedDoor-Fixed").

## 3.3 LIMITATIONS

In some environments with a big number of different objects our method struggles to converge. The tabular agent does not have the observational invariances that methods based on function approximation learn and exploit (e.g. with CNNs). For instance, for these environments learned invariances allow them to handle differently colored objects. While the tabular agent does cope with avoiding redundancy through the cyclophobic intrinsic reward and the hierarchical state representations, it does not learn the kind of observational invariances neural networks do, which are sometimes necessary. However, note that we can show that reducing the number of colors for the "KeyCorridor" environment improves our performance dramatically (see Appendix A.2). Color reduction can also be seen as a cropped view of the real situation, and hence this experiment falls into our setup. Furthermore, preliminary results with a PPO agent (see Appendix A.1) solves "KeyCorridor", further supporting our theory.

Nonetheless, we argue that the above limitation does not lessen the impact of our work, since the tabular agent is able to compete with neural network based methods showing that exploration does not seem to rely solely on representation learning, but also on clever avoidance of repetition. Thus, when the number of different objects increases, better performance can be attributed to the learned

invariances in a neural network reducing observational complexity, rather than to more efficient exploration.

## 3.4 ABLATION STUDY

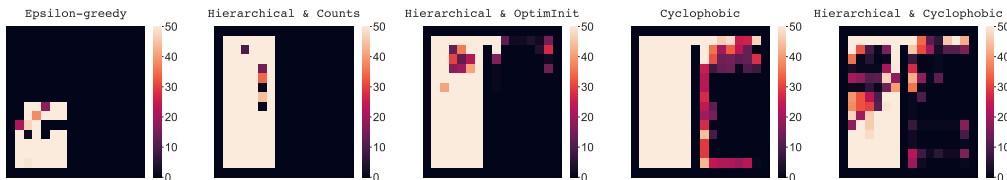

Figure 5: **MiniGrid ablation study (visitation counts for "DoorKey-16x16"):** The cyclophobic Q-function explores the environment more efficiently than the optmistic initialization, count and epsilon-greedy based counterparts. We record visitation counts for several variations of intrinsic reward and hierarchical state representations. Hierarchical state representations together with the cyclophobic intrinsic reward are the most efficient.

To show the effectiveness of the cyclophobic intrinsic reward and the hierarchical state representations, we perform an ablation study. Figure 5 shows state counts as a heat map after 10,000 steps of training. We distinguish four cases: **(i) epsilon-greedy:** Plain epsilon greedy exploration fails to find the goal in the bottom right. **(ii) hierarchical & counts:** We replace the cyclophobic intrinsic reward in equation (3) with a count-based intrinsic reward similar to Strehl & Littman (2008) defined by $N(o_T^{V_i})^{-\frac{1}{2}}$, for view $V_i$. This improves the results but still fails. **(iii) hierarchical & optimistic initialization:** We use optimistic initialization to let the agent try all possible actions. We initialize the q-table with a value of two. Optimistic initialization manages to enter the second room but just fails at getting a reward within 10,000 steps. **(iv) cyclophobic:** Having a cyclophobic intrinsic reward calculated only on the largest view finds the goal. **(v) cyclophobic & hierarchical:** The combination of hierarchical views and cyclophobic intrinsic rewards (as explained in Sec. 2.3) explores even more efficiently as can be seen in the left room where fewer steps are needed to leave it.

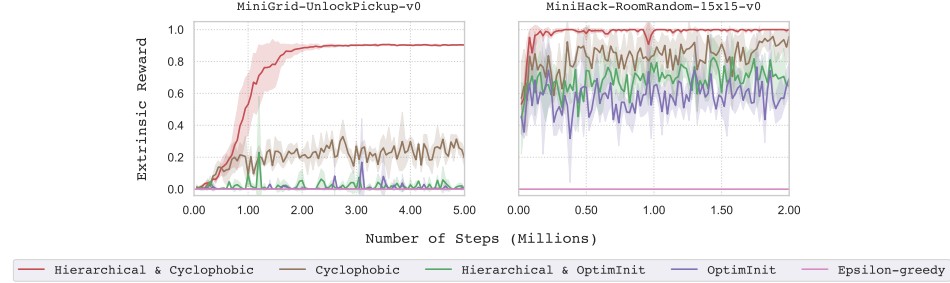

Figure 6: **MiniGrid ablation study (extrinsic reward):** Hierarchical state representations are crucial to performance in some environments. We see that the cyclophobic intrinsic reward is the only method that can solve the environments. Optmistic initialization finds succesful trajectories, but is not as consistent.

In Figure 6 we analyze the impact of the hierarchical state representations on the agent's performance. The training setup is described in section C.3. The "UnlockPickup" environment can only be solved if hierarchical state representations and the cyclophobic intrinsic reward are used. Due to the complexity of the environment and the large number of object variations, the different views allow for knowledge reuse and thus lead to better performance. Similarly, for the MiniHack "Room-Random" environment the cyclophobic intrinsic reward performs well and adding hierarchical state representations leads to the environment being solved. Optimistic Initialization is able to collect some rewards in both environments and the hierarchical state representation improve performance. However, performance still falls short to cyclophobic reinforcement learning.

## 4  RELATED WORK

**Systematic exploration and loop closure:**   Systematic exploration similar to that of the cyclophobic intrinsic reward has been previously explored in the form of optimistic initialization. The idea of optimistic initialization is to initialize the value function by a value larger than the expected reward. As the agent performs updates on the value function it will be forced to choose the next unexplored action since the update for the current state-action pair will decrease the current value. Machado et al. (2014) suggest normalizing the reward by the first encountered extrinsic reward and then subtracting one. In this way the value function can be initialized with zero and all further updates provide a negative value, leading the agent to explorative behaviour as in regular optimistic initialization. Mirowski et al. (2016) define loop closure as an auxiliary loss for exploration in 3D mazes in addition to a depth measure. The authors detect loop closure by checking whether the agent has been near a previously visited position. To get position information the authors integrate velocity information over time.

**Exploration by counting changes in observations:**   Instead of relying on the prediction error to measure state transition change, the changes in the observation space can be counted. Zhang et al. (2020) define an intrinsic reward for a count based approach by calculating the difference of visitation counts between subsequent states. This intrinsic reward pushes the exploration boundary as it provides a positive reward when reaching unexplored states for the first time or no reward otherwise. This mechanism to check novelty is extended through NovelD(Zhang et al., 2021). Interesting Object, Curious Agent by Parisi et al. (2021) is an intrinsic motivation method that splits up learning into a pretraining and transfer stage. It defines a count-based intrinsic reward called C-BET that is based on counting the state transition changes $N(c(s, s'))$. In contrast, we use state counts to measure novelty in the respective hierarchical views.

**Encoding observations to augment intrinsic reward:**   Encoding observations to either augment the intrinsic reward or to define goals has lead to great improvements in exploration. Language in the form of text prompts that define goals has been used by Mu et al. (2022) to augment the intrinsic reward. Singh Chaplot et al. (2020) build representations that reflect the structure, i.e. geometry of the environment. More generally, Ye et al. (2020) train an A2C (Mnih et al., 2016) agent on different croppings in a grid environment, without an intrinsic reward, with mixed results. Parisi et al. (2021) also augment the input by training on a 360° panoramic to learn task-agnostic changes. Our hierarchical state representations similarly augment the intrinsic reward and reveal structural invariances about the environment which aid exploration.

**Generalization and transfer learning:**   Learning representations of the environment that are reusable and can be transferred has been explored previously. Learning successor representations (Barreto et al., 2016) requires learning local environment dynamics that can be reused when the environment, i.e. the original distribution, is changed. (Schaul et al., 2015) use Singular Value Decomposition on the learned action-value function to obtain a canonical representation of the environment which can be transferred to other situations. More recently, Parisi et al. (2021) learn an exploratory policy which is then combined with a task specific policy at transfer time. Our method likewise contains a task-agnostic component given by the hierarchical state representations and a task-specific component given by the cyclophobic intrinsic reward.

## 5  CONCLUSION

Avoiding cycles allows the agent to quickly and systematically discard uninteresting state/action pairs which are repeated often. This makes our cyclophobic intrinsic reward a good inductive bias towards novelty that simultaneously encourages systematic exploration. The ablation studies show that the cyclophobic intrinsic reward just for a single view is already powerful enough to solve complex environments. Adding hierarchical state representations leads to even better performance, as can be seen in the MiniGrid experiments. Moreover, the experiments in MiniGrid confirm the transferability of the learned bias to new environments. Here the hierarchical state representations are crucial, as otherwise in a tabular approach transferability could not be achieved. Due to the tabular approach, environments with large object variation lead to convergence problems in our method. The more objects are present in a single view, the larger the state space becomes in a tabular setting. Neural networks, on the other hand, learn invariances that reduce the complexity of the state space. To address the convergence problems, future work will therefore focus on incorporating the hierarchical state representations into a neural network based architecture.

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

## A   FURTHER ABLATION STUDIES

For all plots in the ablation studies we use the same setup as in the main experiments. We refer to section C.3 for this.

### A.1   PRELIMINARY RESULTS WITH PPO

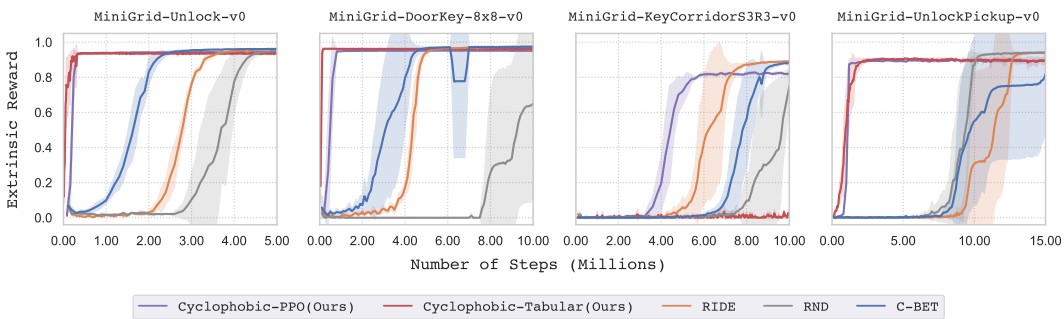

Figure 7: Using PPO together with only the cyclophobic intrinsic reward also converges far quicker than C-BET. Moreover, we are able to converge in KeyCorridorS3R3 confirming our theory in Section 3.3.

We train a PPO agent using only the cyclophobic intrinsic reward. Figure 7 shows that for the given environments the PPO agent also explores more efficiently than C-BET. Furthermore, our theory about observation complexity from 3.3 is supported as the PPO agent is able to solve the "KeyCorridor" environment. Note that these are preliminary results. In future work we seek to implement the hierarchical state representations for the neural agent.

### A.2   OBSERVATIONAL COMPLEXITY - REMOVING COLOR FROM OBSERVATIONS

To test our theory of observational complexity affecting our performance in some environments such as "KeyCorridor" and "ObstructedMaze-2Dlhb", we remove color from objects in the environment such that the underlying task is not affected. In case of "KeyCorridor" and "MultiRoom" colors can be removed from all objects, while in the "ObstructedMaze" environments we only remove color of the key and door. We show the results in figures 8, 9 and 10.

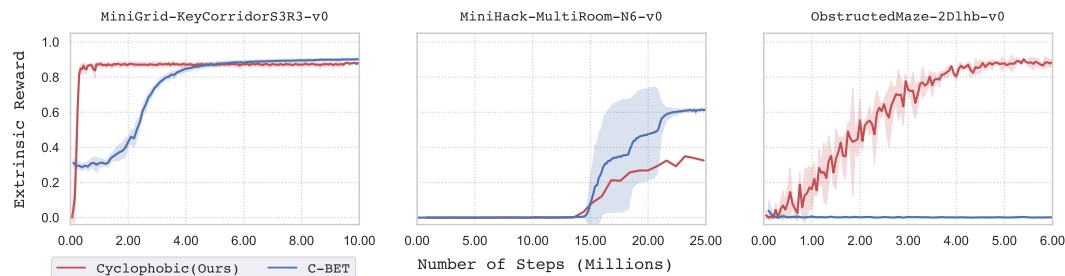

Figure 8: When extracting color from the hierarchical state representations we are able to converge very quickly in the "KeyCorridorS3R3" and "ObstructedMaze-2Dlhb" environments. Doing this for the MultiRoom-N6 environment does not have such a large effect as here the large variation in observation is due to the many different corridors, which our tabular method cannot find observational invariances for as a neural network could.

Figure 8 shows that this has an immediate impact on the performance where the "KeyCorridorS3R3" environment converges instantly while the "ObstructedMaze-2Dlhb" environment needs very few steps to converge in comparison to C-BET. For the MultiRoom-N6 environment, the observational complexity comes from the different types of corridors present throughout training, thus removing color only mitigates the observational complexity to some extent. This corroborates our theory that while our method is very good at exploration, its tabular nature cannot capture the observational invariances that neural networks naturally do.

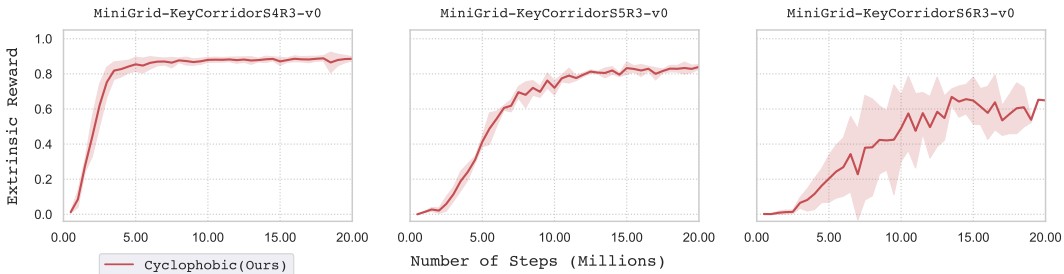

Figure 9: Our agent can even gather rewards successfully in much larger KeyCorridor environments when removing colors, showing that the exploration mechanism is very strong.

In Figure 9 we test more difficult "KeyCorridor" environments. We can see that the cyclophobic reinforcement learning is also able to explore efficiently in these harder environments when colors are removed.

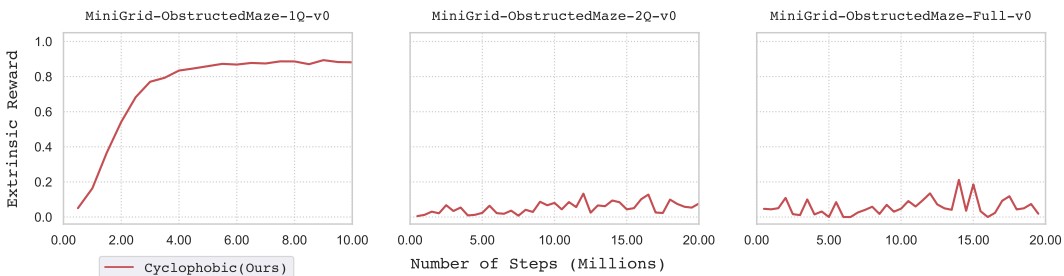

Figure 10: For the more difficult "ObstructedMaze" environments the agent is able to solve the "1Q" version. The other more larger variants prove too complex even when removing colors.

Finally, we test more complex "ObstructedMaze" environments in Figure 10. While we achieve good performance in the "1Q" variant, the other larger variants remain too complex for the cyclophobic agent.

Note that when training the other methods without colors their performance may improve. However, this ablation study simply illustrates that our method is very good at exploring and is only held back by its tabular nature. In future work we want to find the best way to learn observational invariances with the hierarchical state representations and the cyclophobic intrinsic reward.

### A.3 COMPARISON WITH MORE RECENT METHODS

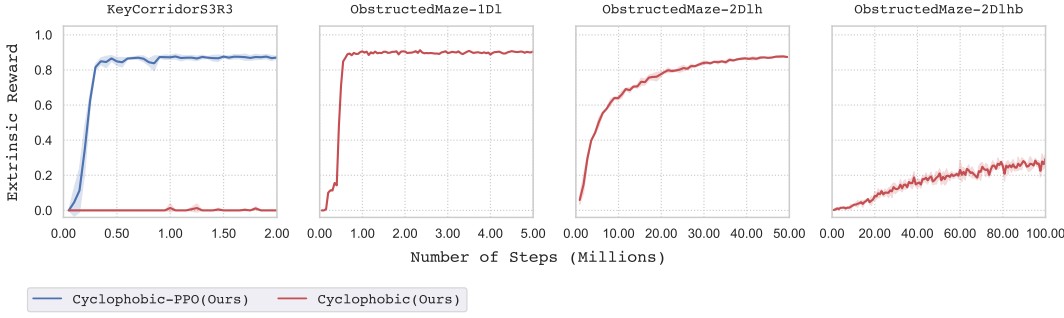

Figure 11: Selection of environments that overlap with more recent methods such as Zhang et al. (2021), Campero et al. (2020), Zhang et al. (2021)

# B ENVIRONMENT DETAILS

## B.1 MINIGRID EXPERIMENTS

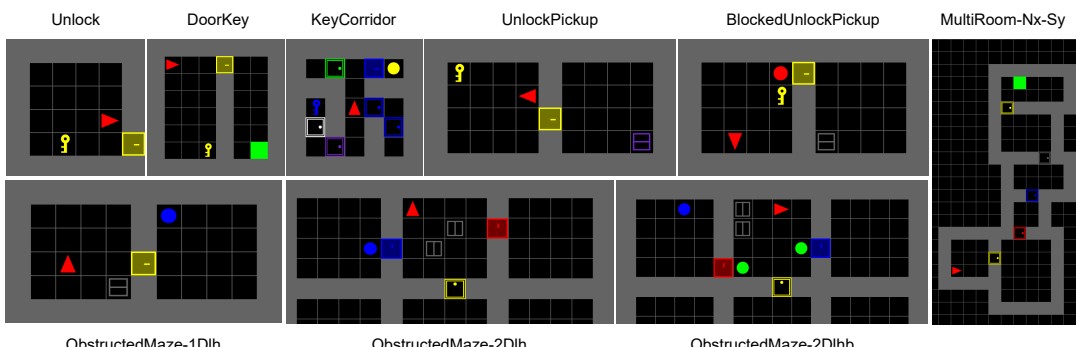

Figure 12: An overview of the MiniGrid environments used in this paper.

We test on some of the same environments as Parisi et al. (2021). In the following we describe the environments as Parisi et al. (2021) and add our own comments where relevant.

- **Unlock**: pick up the key and unlock the door. (288 steps per episode)
- **DoorKey-8x8:** pick up the key, unlock the door, and go to green goal (640 steps per episode)
- **KeyCorridorS3R3:** pick up the key, unlock the door, and pick up the ball (only the door before the ball is locked) (270 steps per episode).
- **UnlockPickup:** pick up the key, unlock the door, and open the box (288 steps per episode).
- **BlockedUnlockPickup:** pick up the ball in front of the door, drop it somewhere else, pick up the key, unlock the door, and open the box (576 steps per episode).
- **ObstructedMaze-1Dlh:** open the box to reveal the key, pick it up, unlock the door, and pick up the ball (288 steps per episode).
- **ObstructedMaze-2Dlh:** same as above, but with two doors to unlock (576 steps per episode).
- **ObstructedMaze-2Dlhb:** same as above, but with two balls in front of the doors (like in BlockedUnblockPickup) (576 steps per episode).
- **MultiRoom-N6:** navigate through six rooms of maximum size ten and go to the green goal (all doors are already unlocked) (120 steps per episode).
- **MultiRoom-N12-S10:** navigate through twelve rooms of maximum size ten and go to the green goal (all doors are already unlocked) (120 steps per episode).

## B.2 MINIHACK EXPERIMENTS

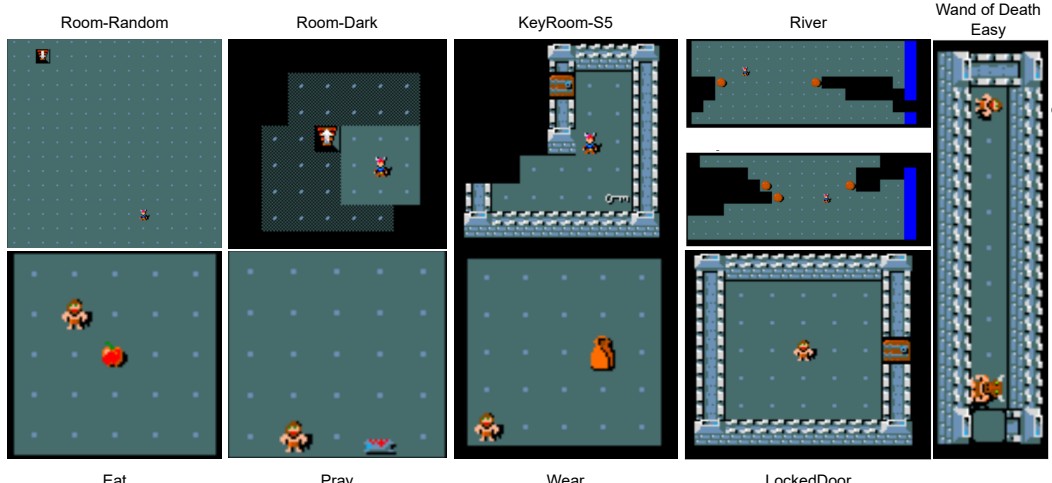

Figure 13: An overview of the Minihack environments tested in this paper. The top row shows the *Navigation* tasks, while the bottom row shows the *Skill* tasks.

We perform experiments inspired by the baselines from Samvelyan et al. (2021). Here, Samvelyan et al. (2021) define a set of navigation and skill acquisition tasks. The navigation tasks range from requiring the agent to navigate an empty room or maze, up to navigating environments with monsters and hazards. The skill acquisition tasks require the agent to perform a certain set of skills to be able to solve the environments.

### B.2.1 NAVIGATION TASKS

In the following we describe the environments used from the navigation task.

- **Room-Random-15x15:** This task is set in a single square room, where the goal is to reach the staircase down. In the random version of this task, the start and goal positions are randomized.
- **Room-Dark-15x15:** Same as Room-Random, but the agent only sees parts of the environment it has already discovered.
- **KeyRoom-S5:** These tasks require an agent to pick up a key, navigate to a door, and use the key to unlock the door, reaching the staircase down within the locked room. The locations of the door, key and starting position are randomized.
- **River:** The river environment requires the agent to cross a river using boulders. When pushed into the water, they create a dry land to walk on, allowing the agent to cross it.

### B.2.2 SKILL TASKS

The nature of commands in NetHack requires the agent to perform a sequence of actions so that the initial action, which is meant for interaction with an object, has an effect. The exact sequence of subsequent can be inferred by the in-game message bar prompts. Overall, in each of the following tasks the agent has to perform a sequence of actions that allow the agent to perform the required skill.

- **Eat:** The agent has to navigate and eat an apple. Doing so requires a sequence of actions and confirming a prompt to eat the appe.
- **Pray:** The agent has to pray on an altar. This is more complex than eating an apple as praying requires the agent to confirm more prompts.
- **Wear:** The agent has to wear armor. This is a more difficult task than the previous tasks.
- **LockedDoor-Fixed:** The agent has to kick a locked door.
- **Wand of Death-Easy:** The agent starts with a wand of deatch in its inventory and has to zap it towards a sleeping monster. That is, the agent needs to equip the wand of death walk to the monster and zap it.

In all of these environments the positions of the agent and of the object that has to be interacted with are randomised. Furthermore, the complexity is also increased by the confirmation prompts the agent has to answer in order to perform a certain action.

## C  EXPERIMENTAL DETAILS

### C.1  HYPERPARAMETERS

**Common hyperparameters:**  In the following we present a table with all hyperparameters used for training the environments. In general, our method has very few hyperparameters. In this case the only really tunable hyperparameters are the step size $\eta$, the random action selection coefficient $\epsilon$ for epsilon-greedy exploration and the discount $\gamma$:

| $\eta$ | $\gamma$ | Num. views |
|---|---|---|
| 0.2 | 0.99 | 5 |

In our case we determined experimentally that 5 views is a good number of views for training an agent.

**Epsilon-greedy parameter $\epsilon$:**  While learning with the cyclophobic intrinsic reward we still perform random action from time to time by having an epsilon-greedy action selection. That is, the agent samples a value $n$ from a uniform distribution and selects a random action if this $n < \epsilon$. These are the settings for $\epsilon$ for the respective environments:

| Environments | $\epsilon$ |
|---|---|
| Unlock | 0.1 |
| DoorKey | 0.1 |
| KeyCorridor | 0.1 |
| UnlockPickup | 0.3 |
| MultiRoom-N6 | 0.1 |
| MultiRoom-N12-S10 | 0.1 |
| BlockedUnlockPickup | 0.3 |
| ObstructedMaze-1Dlh | 0.3 |
| ObstructedMaze-2Dlh | 0.1 |
| ObstructedMaze-2Dlhb | 0.3 |

**Intrinsic reward tradeoff $\rho$:**  For the "ObstructedMaze-2Dlh" and "BlockedUnlockPickup" environments we set intrinsic reward tradeoff $\rho = 0.2$. This has as effect that the extrinsic reward gets propagated more when the agent encounters it, which is necessary for more complex environments. For all other tested environments we set $\rho = 1$, thus not weighting the intrinsic reward at all.

### C.2  PPO HYPERPARAMETERS

| Intrinsic reward tradeoff $\rho$ | 0.01 |
|---|---|
| GAE-$lambda$ | 0.95 |
| $\gamma$ | 0.99 |
| Batch size | 256 |
| Learning rate | 0.001 |
| Entropy coefficient | 0.01 |
| Value loss coefficient | 0.5 |
| Max Grad Normt | 0.5 |
| Clipping-$\epsilon$ | 0.2 |
| RMSProp-$\epsilon$ | 1e-8 |
| RMSProp-$\alpha$ | 0.99 |

### C.3  PLOT SMOOTHING

For all 3 runs we show the mean reward and standard deviation smoothed over a sliding window. For our and Samvelyan et al. (2021)'s experiments we use a sliding window of 50000 steps. For the experiments in Parisi et al. (2021) we use a sliding window of 96000 steps as per their experimental setup. The shaded areas are the standard deviation of the mean for 3 runs.

