# OpenReview forum: "Cyclophobic Reinforcement Learning"
_ICLR.cc/2023/Conference — Submitted to ICLR 2023_

### Official Review · Reviewer_CKV7 · 2022-10-24

**Confidence:** 4
**Correctness:** 2
**Technical Novelty And Significance:** 2
**Empirical Novelty And Significance:** 2
**Recommendation:** 3

**Clarity, Quality, Novelty And Reproducibility:**

The paper is clearly written. However, the experiment section is unsound and I'm concerned about the reproducibility of the reported results. The novelty can be also limited (see my comments above).

**Strength And Weaknesses:**

## Strength
This paper is clearly written and easy to follow. The loop-closure-based intrinsic reward is intuitive and the method is easy to implement. The idea of building a hierarchy of patches for loop detecting is novel. Experiments are conducted on two challenging experiments, which is good.


## Weakness

My concerns are mainly about novelty and experiments, which are listed below.

1. **Limited novelty**: The idea of detecting loops to encourage exploration isn't really a novel idea. Its practice in deep RL literature can be at least traced back to [1], which implements an additional loss to predict loops. Even the idea of using loop-based rewards is well executed as a specific technique in the NoveID method [2]. The hierarchy of patches for loop detection is interesting and could be counted as a great practical technique for realizing loop detection. However, I would be concerned about whether the paper contributes enough to the community, particularly based on the limited experiments (see my following comments).

[1] Learning to Navigate in Complex Environments, Piotr Mirowski et. al, ICLR 2017

[2] NovelD: A Simple yet Effective Exploration Criterion, Tianjun Zhang, Huazhe Xu, Xiaolong Wang, Yi Wu, Kurt Keutzer, Joseph E. Gonzalez, Yuandong Tian, NeurIPS 2021.

2. **Missing baselines**: The paper primarily compares with two baselines, i.e., RIDE and RND. However, many recently developed baselines are omitted. More importantly, many of them are also tested on the miniGrid environment. Representative examples include NoveID [2], AMiGo [3], MADE [4] etc.

[3] Learning with AMIGo: Adversarially Motivated Intrinsic Goals, Andres Campero, Roberta Raileanu, Heinrich Kuttler, Joshua B. Tenenbaum, Tim Rocktäschel, Edward Grefenstette, ICLR 2021

[4] MADE: Exploration via Maximizing Deviation from Explored Regions, Tianjun Zhang, Paria Rashidinejad, Jiantao Jiao, Yuandong Tian, Joseph E. Gonzalez, Stuart Russell, NeurIPS 2021.

3. **Experiment results seem unfinished**. It is particularly wired to me that there are no reward curves at all for the MultiRoom-N12-S10 scenario in Fig 2. At least, if you check the RIDE paper, RIDE can definitely achieve non-zero rewards by training from scratch.



**Summary Of The Paper:**

This paper proposes an intrinsic reward for RL exploration based on loop-closure detecting. When a state or a central patch of the state is visited, a penalty will be given to the agent to encourage the agents to visit novel states. Such an intrinsic reward design is combined with Q-learning to conduct experiments on MiniGrid and MiniHack environments.

**Summary Of The Review:**

On the technical part, the idea of building a hierarchy of patches for developing a set of intrinsic rewards is interesting. However, the contribution is not well supported by the experiments, which further makes the novelty of this work limited.

---

> ### Author Response · Authors · 2022-11-15
> **Response to review comment**
>
> We want to thank the reviewer for their detailed review. In the following, we will answer your questions and hope you might raise your score.
>
> > The idea of detecting loops to encourage exploration isn't really a novel idea. Its practice in deep RL literature can be at least traced back to [1], which implements an additional loss to predict loops.
>
> Thanks for the early reference [1], which we added to the related work section.  However, we note, the cyclophobic intrinsic reward can be seen as a combination of optimistic initialization and loop closure detection. Penalizing the cycles makes the agent try (almost) every action for every state while being able to discard unpromising state-action pairs fairly quicky thanks to the loop detection.  This is different from the existing literature.
>
>
> > Even the idea of using loop-based rewards is well executed as a specific technique in the NoveID method [2].
>
> Thanks for the reference.  However, [2] is a bit different to loop detection, since they only give a reward when visiting a boundary state the first time and not for subsequent visits. Our method in contrast gives a penalty everytime a loop is detected.
>
> > The hierarchy of patches for loop detection is interesting and could be counted as a great practical technique for realizing loop detection. However, I would be concerned about whether the paper contributes enough to the community, particularly based on the limited experiments (see my following comments).
>
> We show performance that goes way beyond the excellent performance of Parisi et al. (2021), which we consider a big achievement.  Let's restate our contributions more clearly:
>
> 1. The cyclophobic intrinsic reward can be seen as a combination of optimistic initialization and loop closure detection. Penalizing the cycles makes the agent try (almost) every action for every state while being able to discard unpromising state-action pairs fairly quicky thanks to the loop detection.
>
> 2. We show that using hierarchical state representations, naturally allows for transfer learning in the tabular case.
>
> 3. Taking this into account, in our limitations section and the appendix, we also show that exploration has both an observational component and a task complexity component. We show this by reducing the number of colors in the observations for MiniGrid showing that our tabular method becomes just as competitive as neural network based methods when the observational complexity is reduced. Conversely, RIDE and RND do not perform as strongly in MiniHack where the task complexity far outweighs the observational complexity
>
> > Missing baselines: The paper primarily compares with two baselines, i.e., RIDE and RND. However, many recently developed baselines are omitted. More importantly, many of them are also tested on the miniGrid environment. Representative examples include NoveID [2], AMiGo [3], MADE [4] etc.
>
> * We primarly compare our method to Parisi et al. (2021) since only they have a transfer learning setup and use a task-agnostic exploration component (panoramic views), which in our case are the hierarchical state representations. So most likely your critique applies to Parisi et al. (2021) as well?
> * NovelD, MADE and AMIGO mostly train on large environments and do not have a transfer learning setup nor a task-agnostic exploration component.
>
> > Experiment results seem unfinished. It is particularly wired to me that there are no reward curves at all for the MultiRoom-N12-S10 scenario in Fig 2. At least, if you check the RIDE paper, RIDE can definitely achieve non-zero rewards by training from scratch.
>
> This is explained in Parisi et al. (2021). We refer to their comment:
>
> > "First, we want to highlight that we have used both the MiniGrid and RIDE baseline directly from their released code (no changes to code). We believe the key reason for the difference is RIDE’s sensitivity to hyperparameters. RIDE needed differently tuned coefficients even for different versions of the MultiRoom. In the original paper, they used tuned intrinsic reward coefficients for each environment. Other baselines (RND, Count, Curiosity) all used the same coefficient, but RIDE performed differently depending on the coefficient.mIn our paper, we used an intrinsic reward coefficient of 0.1 for RIDE. In its original paper, this coefficient is used for MultiRoomN7S4, MultiRoomN10S4, KeyCorridorS3R3. For MultiRoomN7S8, MultiRoomN10S10, MultiRoomN12S10, and ObstructedMaze2Dlh it used 0.5. Finally, we highlight that C-BET worked well in all environments with the same coefficient"
>
> -- Parisi et al. https://openreview.net/forum?id=knKJgksd7kA&noteId=CCYySUbeWbz
>
>
> #### References:
>
> Simone Parisi, Victoria Dean, Deepak Pathak, and Abhinav Gupta. Interesting Object, Curious Agent: Learning Task-Agnostic Exploration. arXiv e-prints, art. arXiv:2111.13119, November 2021.

---

> > ### Comment · Reviewer_CKV7 · 2022-11-16
> > **My disagreement**
> >
> > It is okay to say that your work follows the setting from Parisi et al. (2021). However, at least from your writing before the experiment section, the paper looks like a standard exploration algorithm without any specific algorithmic component related to transfer. Parisi et al. (2021), from the very beginning, emphasize that the work focuses on first learning exploration strategy and then transferring to a new domain.  However, I couldn't see why your algorithm is any special to this transfer setting. It looks like the transfer learning experiment is simply taken as one additional result to consolidate your work. I disagree with the argument that my complaint applies to Parisi et al. (2021). Also, if you really claim that you are trying to follow the setting in Parisi et al. (2021), why not try the Habitat testbed? It is tested by Parisi et al. (2021).
> >
> > It is true that neither Amigo, nor NovelD nor MADE considers the transfer setting. However, they consider the learn-from-scratch setting. I think it remains a valid suggestion that these baselines should be included in the learn-from-scratch part.

---

> > > ### Author Response · Authors · 2022-11-18
> > > **Reply to comment "My disagreement"**
> > >
> > >
> > > We have gathered a series of experiments from the other papers that overlap with ours in minigrid (see Sec. A.3). For tabular methods we are definitely SOTA and beat other (not all) neural network based methods. In larger environments it becomes more difficult for our tabular method to compete due to observational complexity (see Sec. 3.3). More details:
> > >
> > > * **AMIGO**: For ObstructedMaze-1Dl our method converges quicker and more reliably. For KeyCorridor S3R3, while we do not converge with the tabular method the neural network method converges quicker. For ObstructedMaze-2Dlhb we start converging orders of magnitude quicker although performance does not improve after that.
> > >
> > > * **MADE**: ObstructedMaze2Dlh and Obstructed2Dlhb, our method starts to converge quicker less top end performance. For KeyCorridor S3R3 about the same performance.
> > >
> > > * **NoveID**: ObstructedMaze2Dlh and Obstructed2Dlhb, our method starts to converge quicker less top end performance. For KeyCorridor S3R3 about the same performance.
> > >
> > >
> > > We think testing on tabular methods is important as it shows that exploration methods must both account for observational complexity and task complexity (see also Sec.3.3).
> > >
> > > For instance, Parisi et al. show in their experiments that even simple count based exploration works well with neural networks.
> > >
> > > Furthermore, to support Sec. 3.3,  when we train on only one color in the ObstructedMaze and KeyCorridor environments we are much faster than other methods (see Figure 8). Also in Figure 9 we see that the size of the environment becomes critical for the larges ObstructedMazes.
> > >
> > > ----------
> > >
> > > > However, the experiment section is unsound and I'm concerned about the reproducibility of the reported results.
> > >
> > > It is valid to request MADE, AMIGO and noveID, however given the standpoint of our paper (see answer to other questions) and our method being tabular we disagree with the experiments being unsound. Furthermore, the code will be publicly availabe and we now show error margins for all our results.
> > >
> > >
> > > ----------
> > >
> > > > However, I couldn't see why your algorithm is any special to this transfer setting.
> > >
> > > The transfer learning component of our method are the cyclophobic Q-functions learned for the different views (see Sec. 2.3), that learn general properties of MiniGrid like environments.  As explained in "Training setup and baselines" in Section 3, we "save the extrinsic rewards in a second separate Q-table for each view in addition to the main Q-table which contains values of equation 2".
> > >
> > > >Also, if you really claim that you are trying to follow the setting in Parisi et al. (2021), why not try the Habitat testbed?
> > >
> > > Habitat seems to be used by Parisi et al. (2021) to test their panoramic views, which one could understand why it would be a good setting to be used in. We don't use panoramic views hence we didn't include it into our work.

---

### Official Review · Reviewer_FEf9 · 2022-10-25

**Confidence:** 4
**Correctness:** 3
**Technical Novelty And Significance:** 3
**Empirical Novelty And Significance:** 2
**Recommendation:** 3

**Clarity, Quality, Novelty And Reproducibility:**

I found the paper well written and easy to understand.
The various plots and visualization were really helpful to understand the proposed method.

**Strength And Weaknesses:**

Strengths:

_ The paper is well written and easy to follow.
_ I liked the empirical evaluation, it was done on many tasks from MiniHack and MiniGrid with a large set of baselines. The ablation study was also helpful to understand the impact of each component of the method.
_ I also appreciate that the authors discuss the limitations of their method.
_ Results on minigrid and MiniHack are impressive and clearly the benefits of cyclophobic RL over exisiting exploration techniques.

Weaknesses:

_ Overall it is hard to know how the proposed will scale to more complex problems like Atari games where the same state may not be seen multiple times (at least on the largest view). On the hand the narrow may not have enough local information to be useful. The counts of the observations might also be difficult to compute in that situation.
_ From 1 (b) it looks like to obtain a smaller view a rotation is done in addition to cropping? Is that part necessary? How does it impact the performance of the algorithm. Seems like to do the orientation of the agent must be known.

**Summary Of The Paper:**

This paper introduces cyclophobic reinforcement learning, a negative intrinsic reward for exploration that incentivizes the agent not to revisit previously visited states. This intrinsic reward is computed over different cropped views of the environment by the agent and these are combined to compute the final reward. The proposed method is evaluate on mini grid and mini hack and is shown to lead to increased performance.

**Summary Of The Review:**

Despite impressive results on MiniGrid and MiniHack, it is unclear how the proposed method scale to larger environment where it is not possible to count observations. Because of that I am not too sure what we can learn from cyclophobic RL as it is presented in the paper and for that reason I cannot recommend acceptance.

---

> ### Author Response · Authors · 2022-11-15
> **Response to review comment**
>
> We thank the reviewer for their detailed review. We hope we are able to provide more insights how the method could be scaled to larger and more complex environments.  Of course, if we convince you, we would be happy if you raised your score.
>
> > Overall it is hard to know how the proposed will scale to more complex problems like Atari games where the same state may not be seen multiple times (at least on the largest view). On the other hand the narrow may not have enough local information to be useful. The counts of the observations might also be difficult to compute in that situation.
>
> MiniGrid and MiniHack are environments which were designed to be hard to explore, so they are the common benchmark for methods trying to improve exploration with with sparse rewards, see e.g.,  Parisi et al. (2021) .  However, performance on Atari would be interesting, but is beyond the current scope of our paper, which focusses on comparisons against other methods, that have been mostly evaluated on sparse rewards such as MiniGrid.
>
> > From 1 (b) it looks like to obtain a smaller view a rotation is done in addition to cropping? Is that part necessary? How does it impact the performance of the algorithm. Seems like to do the orientation of the agent must be known.
>
> We use the default implementation of MiniGrid which automatically orients the agent, exactly like the experiments in other paper, e.g.,  Parisi et al. (2021).
>
> MiniHack does not have this rotation but still performs well, which shows that the orientation is not even essential to our method.
>
>
> #### References:
>
> Simone Parisi, Victoria Dean, Deepak Pathak, and Abhinav Gupta. Interesting Object, Curious Agent: Learning Task-Agnostic Exploration. arXiv e-prints, art. arXiv:2111.13119, November 2021.

---

### Official Review · Reviewer_FYSy · 2022-10-26

**Confidence:** 3
**Correctness:** 3
**Technical Novelty And Significance:** 2
**Empirical Novelty And Significance:** 2
**Recommendation:** 5

**Clarity, Quality, Novelty And Reproducibility:**

While I felt the paper was reasonably clear and well-structured, there were some areas that I felt were not sufficiently clear. In particular, I don't understand how the transfer learning across different environments (pretraining) was achieved. A couple of areas I found especially confusing:
1. "We simply save the extrinsic rewards in a separate Q-table and set the values as starting values for the new Q-table at the start of transfer learning." (p. 5) ← I don't understand what this means. Which extrinsic rewards does this refer to and which values does this refer to?
2. "Our method likewise contains a task-agnostic component given by the cyclophobic intrinsic reward and a task-specific component given by the external reward." (p. 9) ← I don't understand this part; are the visitation counts for different views re-used in multiple environments?

While I feel like I have a sufficient understanding of how cycles fit into the use of hierarchical views, I didn't always follow the explanations and examples as written; writing those parts in another way could really improve the clarity of the paper.
1. "encountering a key in a smaller view produces less cycles" (p. 1) ← I don't follow this example
2. "Surprisingly, in combination with the cyclophobic intrinsic reward, we gain additional information about the structure of the environment." (p. 2) ← I don't understand this statement.
3. "the most detailed view" (p. 3) ← I think this is referring to the global view (has the most details) but it could be referring to the most local view (most zoomed in on the details)

Other areas of confusion for me:
- "will propagate the additional penalty across the trajectory" (p. 2) → I would benefit from more explanation of the propagation
- "That is, where some form of disentangled representation of the environments plays an important role for efficiently finding solutions." (p. 5) ← for a reader who doesn't know what the binding problem is (whom I assume this statement is for) this statement doesn't help.
- "We train each environment" (p. 5) ← I'm guessing you didn't mean to say environment here, probably meant "agent"? Can you speak more about what you mean by "train" in this paper? I believe the "no transfer" experiments do not involve distinct training and testing phases, so I'm guessing that when you talk about training, you are referring to all agent learning and behaviour?
- "which are diverse in the required skills." (p. 6) ← not sure what this means

Reproducibility concerns:
1. The ablation study is missing a number of details. Did you perform multiple runs? Which MiniGrid environment was used to generate Figure 5?
2. How was the shaded area in the figures computed?

**Strength And Weaknesses:**

Cyclophobic rewards are an intuitive method for encouraging systematic exploration in sparse reward environments. Part of what makes cyclophobic rewards intuitive is that they penalize states that have been visited before, encouraging the agent to find parts of the world it hasn't visited before to avoid such penalties. This intuition is similar to that we apply to optimistic initialization—the states the agent visits are worse than it expects, so the agent is encouraged to try elsewhere. In fact, there is a method that uses negative intrinsic rewards to create the effect of optimistic initialization (Machado, Srinivasan & Bowling, 2015).

Without understanding how or whether the effects of cyclophobic rewards extend beyond the typical effects of other mostly negative intrinsic rewards, it is difficult to evaluate the significance of cyclophobic rewards. To the best of my understanding, all of the baseline intrinsic reward methods used for comparison in this paper use non-negative intrinsic rewards.

My other concern with the baselines is the state representation used. In Section 3.1, it says that the "training setup of Parisi et al. (2021)" is used, but I wasn't sure which aspects of the training setup were meant. I believe Parisi et al. used a panoramic state view that obscures anything behind walls or closed doors, but from Figure 1 in this paper, I get the impression that the agent's state representation includes the whole gridworld, including items behind walls—can you be more explicit about what state information the agents had access to in the experiments? Did all of the agents have access to the same state information? (If so, I'm guessing we should expect different results than those presented by Parisi et al., 2021.)

I found that the structure of the paper was straightforward and easy to follow. The paper is well-written, for the most part. I appreciated how the authors anticipated many of my concerns throughout, and noted limitations accordingly. However, another concern I don't feel was addressed is the reasonableness of maintaining state histories for multiple views. MiniGrid environments are relatively small; do you feel that this method will be applicable as environments get larger?

The related work components represent one of the weaker parts of the paper; I did not find the relationships between the proposed method and the existing work very clearly explained.
1. "One way to do this is by approximating the counts via a density model as done by Ostrovski et al. (2017) and Bellemare et al. (2016)." (p. 8) Given that your algorithm as described is tabular, I don't see the relevance of the translation to pseudo-counts.
2. Do you consider RND and RIDE to be at all related to your method, or are they just described in the related work (p. 9) because they are included baselines in some of the experiments?
3. "Overall, prediction error methods require a model while our method is model-free." (p. 9) ← This statement is a little strange coming shortly after your summary of RND, which doesn't require a model in the traditional sense.

Minor error that should be fixed:
- "We see that the cyclophobic intrinsic reward is necessary to reach a successful trajectory at all." (p. 8) Not sure what this means, since the figure only shows a comparison of two versions of cyclophobic agents against epsilon-greedy. Probably should be rephrased.

Typos and grammatical suggestions (no intended impact on score)
- "In environments with sparse rewards finding" (p. 1) → "In environments with sparse rewards, finding"
- "Random exploration (e.g. epsilon-greedy with sparse rewards)" (p. 1) ← Makes it sound like sparse rewards are part of the exploration method rather than the environment—and isn't this still a problem even when rewards are dense?
- "This idea is further pushed by" (p. 1) ← not sure what this means; may want to rephrase
- "like avoid everywhere." (p, 1) → "like to avoid everywhere."
- "Instead rewarding novelty we" (p. 1) → "Instead of rewarding novelty, we"
- "where an observation function Z : S → O maps the observation to the true state." (p. 2) ← The description is mixed up; Z maps the true state to the observation that the agent gets.
- "the cycle penalty is −l. That is, during exploration the cycle penalty can decrease indefinitely." (p. 2) ← This is a little awkward because if the penalty is decreasing, it sounds like it should be getting less negative. There are a few alternatives like referring to it as "the cyclophobic reward" or talking about the magnitude of the penalty increasing indefinitely, but I haven't thought of a way to completely avoid confusion.
- "how it can be build" (p. 2) → "how it can be built"
- "is the sum of" (p. 2) → might be better to say "is the weighted sum of" to account for rho.
- "restricting the views leads to" (p. 2) ← "restricting the view leads to" might be a little clearer
- "V_1 is", "V_2 are", "V_3 is" (p. 3) ← probably want to be consistent
- "what is front" (p. 3) → "what is immediately in front of it"
- "to V_k the" (Figure 1 caption) ← missing space?
- "several weights" (p. 4) → There is one weight for each view, is that right? The word "several" is a little vague, and explaining where the selection of weights comes from would help solidify understanding.
- "the next paragraph." (p. 4) ← "the next section"?
- "hashcode" or "hash code" (p. 5) ← probably want to be consistent
- "where m_t \in R^k be" (p. 5) → "where m_t \in R^k is"
- "considered to be solved, if one" (p. 5) → "considered solved if one"
- "our x-axis is shorter" (p. 5) ← Shorter than what? Is this in comparison to the figures shown by Parisi et al. (2021)?
- "dificult" (p. 6) → "difficult"
- "tasks show in Figure 3" (p. 6) → "tasks shown in Figure 3"
- "can not" (p. 7) → "cannot"
- "difficult set of" (p. 7) → "difficult sets of"
- "performance as well dramatically" (p. 7) → "performance dramatically"
- "state counts" (pp. 7, 8, 9) → "visitation counts"
- The "count-based intrinsic reward" (p. 8) used for the hierarchical learner with no cyclophobic reward is based on MBIE-EB (Strehl and Littman, 2008), right? A citation to where the choice of reward came from would be helpful context.
- "representations we achieve" (p. 8) → "representations do we achieve"
- "The mechanism to check novelty" (p. 9) → maybe "This mechanism to check novelty"?
- "amount of changes in a state transition" (p. 9) ← Not sure what this means; what is a change in a state transition?
- "experiments for MiniGrid" (p. 9) → "experiments in MiniGrid"?
- If you get the chance to go through and make the colors for each method consistent (e.g. cyclophobic with hierarchical views could be the same blue in every figure), that would make the figures much easier to comprehend.

Machado, M. C., Srinivasan, S., & Bowling, M. (2015). Domain-independent optimistic initialization for reinforcement learning. In AAAI Workshop: Learning for General Competency in Video Games 2015.

Strehl, A. L. and Littman, M. L. (2008). An analysis of model-based interval estimation for Markov decision processes. Journal of Computer and System Sciences, 74(8):1309 – 1331.

**Summary Of The Paper:**

In this paper, the authors present an intrinsic reward based on cyclophobia—avoidance of cycles of experience. The idea is, to focus on observing parts of the world that haven't been seen before, actions that return the agent to an observation it has made before are penalized. In particular, such a cycle does not even have to mean returning to a previously experienced complete state (in the Markov decision process sense)—bumping up against one wall of a gridworld is as pointless as bumping up against another. To avoid these experiences that are locally cyclic, but not globally cyclic, the authors look for cycles in agent-centrically local "views." In the gridworld experiments the authors use, this means one square in front of the agent for the most local view, the squares to the front and sides in the next most local view, and so on and so forth, with the largest view being the entire gridworld—but we can imagine future work considering other representations of locality more appropriate to non-gridworld domains.

The authors demonstrate that their method improves and speeds up learning in comparison to several other well-known methods in two benchmark gridworld domain settings: MiniGrid and MiniHack. They also perform a small ablation study to provide some evidence that hierarchical views and cyclophobic reward work together to improve performance.

**Summary Of The Review:**

I am leaning to reject this paper, with particular concern about how the experiments were set up (questions about state information above). As I have noted above, not comparing or discussing other mostly-negative intrinsic rewards or optimistic initialization means that I also don't know if this paper presents anything significant. However, the paper includes some ideas about focusing on varying levels of locality and an intuitive intrinsic reward method that could support interesting future work, so it is close to acceptable for me. If we don't accept it at this conference, I hope that the authors continue to work on the clarity of the paper and its contextualization with existing methods so it can be a stronger contribution.

---

> ### Author Response · Authors · 2022-11-15
> **Response to review comment**
>
> We want to thank the reviewer for this detailed review. The insight on optimistic initialization is very valuable and will incorporated into the revised paper. In the following, we will answer your questions to the best of our possibilities.
>
> ## General Concerns
>
> > Without understanding how or whether the effects of cyclophobic rewards extend beyond the typical effects of other mostly negative intrinsic rewards, it is difficult to evaluate the significance of cyclophobic rewards. To the best of my understanding, all of the baseline intrinsic reward methods used for comparison in this paper use non-negative intrinsic rewards.
>
> * We will include a comparison with optimistic initialization, since as you pointed out behaves similarly to our approach. However, optimistic initialization is not really an intrinsic reward in the traditional sense.
> * Furthermore, most (all?) state of the art exploration methods use non-negative intrinsic rewards, against which we compare our approach.
>
> > My other concern with the baselines is the state representation used. In Section 3.1, it says that the "training setup of Parisi et al. (2021)" is used, but I wasn't sure which aspects of the training setup were meant. I believe Parisi et al. used a panoramic state view that obscures anything behind walls or closed doors, but from Figure 1 in this paper, I get the impression that the agent's state representation includes the whole gridworld, including items behind walls. Can you be more explicit about what state information the agents had access to in the experiments? Did all of the agents have access to the same state information? (If so, I'm guessing we should expect different results than those presented by Parisi et al., 2021.)
>
> * We use exactly the same setup as Parisi, including the occluded view that Parisi use.  Only for illustration we show whole environment.  We clarified this in Section 3.1.
>
> > However, another concern I don't feel was addressed is the reasonableness of maintaining state histories for multiple views. MiniGrid environments are relatively small; do you feel that this method will be applicable as environments get larger?
>
> * For determining the cycles this isn't a problem since the history is reset episodically. As in C-BET from Parisi et al.
>
> * Counts are maintained for other intrinsic reward methods as well such as Parisi and noveID. In environments where observations are more complex, a hashing algorithm such as SimHash
> could be used to cluster similar observations.  Our method could be implemented around SimHash as well.
>
> > "We simply save the extrinsic rewards in a separate Q-table and set the values as starting values for the new Q-table at the start of transfer learning." (p. 5) ← I don't understand what this means. Which extrinsic rewards does this refer to and which values does this refer to?
>
> * Our method is tabular, thus for the no pretraining approach we save q-values in a single q-table where intrinsic and extrinsic rewards are combined as defined in equation 2.
>
> * When doing transfer learning, we additionally create a second q-table which saves the extrinsic rewards only during pretraining. These extrinsic rewards are then used at the beginning of transfer learning for each environment. We continue to use the cyclophobic intrinsic reward when doing transfer learning.
>
> * We clarified this in Section 3 under "Training setup and baselines".
>
> > "Our method likewise contains a task-agnostic component given by the cyclophobic intrinsic reward and a task-specific component given by the external reward." (p. 9) I don't understand this part; are the visitation counts for different views re-used in multiple environments?
>
> You are right, the **task agnostic component** refers to the **hierarchical state representations** which can be reused over multiple environments such as when doing transfer learning from "Doorkey" or "MultiEnv".  The **task specific component** is the **cyclophobic intrinsic reward** which reveals structural invariances for the different environments while exploring.  We corrected this in the manuscript.
>
>
> ## Reproducibility Concerns
> > The ablation study is missing a number of details. Did you perform multiple runs?
>
> * Now we have performed 3 runs as in the main experiments for the ablation study and updated the plots and text accordingly.
>
> > Which MiniGrid environment was used to generate Figure 5?
>
> For Figure 5 the DoorKey-16x16 environment was used.
>
> > How was the shaded area in the figures computed?
>
> The values shown in the plots are the means of the extrinsic reward with a sliding window of 50000 steps over 3 runs. We then compute the **standard deviation of the means** over these 3 runs.  We added this additional information in the appendix and reference to it in the text.

---

> > ### Author Response · Authors · 2022-11-15
> > **Response to review comment (continued)**
> >
> >
> > ## Related Work
> >
> > > "One way to do this is by approximating the counts via a density model as done by Ostrovski et al. (2017) and Bellemare et al. (2016)." (p. 8) Given that your algorithm as described is tabular, I don't see the relevance of the translation to pseudo-counts.
> >
> > That makes sense, we have removed pseudo counts literature from the related work section.
> >
> > > Do you consider RND and RIDE to be at all related to your method, or are they just described in the related work (p. 9) because they are included baselines in some of the experiments?
> >
> > Yes. They are only related as being a popular approach for generating intrinsic rewards and we wanted to compare against and differentiate our work from those. We have also decided to remove this part from the related work section.
> >
> > > "Overall, prediction error methods require a model while our method is model-free." (p. 9) ← This statement is a little strange coming shortly after your summary of RND, which doesn't require a model in the traditional sense.
> >
> > We agree and see above.
> >
> >
> > ## Unclear sections
> >
> > > "encountering a key in a smaller view produces less cycles" (p. 1)  ← I don't follow this example
> >
> > This was not written clearly, we apologize. We replaced this with:
> > "Similarly, an interesting object such as a key, produces less cycles in a smaller view than some other object (since the key can be interacted with). Thus the probability of picking up the key increases, since other \emph{less interesting} observations produce more cycles (e.g a wall)."
> >
> > > "Surprisingly, in combination with the cyclophobic intrinsic reward, we gain additional information about the structure of the environment." (p. 2)  ← I don't understand this statement.
> >
> > We added some subsentence to clarify this.
> >
> > "Surprisingly, in combination
> > with the cyclophobic intrinsic reward, we gain additional information
> > about the structure of the environment as we get different Q-functions for different POMDP's. "
> >
> > > "the most detailed view" (p. 3) ← I think this is referring to the global view (has the most details) but it could be referring to the most local view (most zoomed in on the details)
> >
> > Thank you for pointing this out. We have corrected this. Here we mean the smallest view i.e. most local view.
> >
> > > "will propagate the additional penalty across the trajectory" (p. 2) → I would benefit from more explanation of the propagation
> >
> > We added some better explanation to the text.
> >
> > More details:
> > * We want the cycle penalty to be propagated so the agent avoids uninteresting parts of the environment and navigates to more interesting parts.
> > * SARSA seems for us the most natural approach, since it bootstraps from the next state and thus propagates the cycle penalty.
> > * For instance, hitting a wall during many different trajectories will discourage that area being visited since the penalty will be propagated many times.
> > * Similarly, an area where a key or door is present will incurr less cycle penalties and then be more interesting for the agent to visit since the penalty is propagated less frequently.
> >
> > > "We train each environment" (p. 5) ← I'm guessing you didn't mean to say environment here, probably meant "agent"?
> >
> > Thank you for pointing out this mistake. It is corrected.
> >
> > > Can you speak more about what you mean by "train" in this paper? I believe the "no transfer" experiments do not involve distinct training and testing phases, so I'm guessing that when you talk about training, you are referring to all agent learning and behaviour?
> >
> > * Yes exactly, in this case we refer to all agent learning and behaviour.
> > * Overall, we have two setups:
> >     1. We train only once on the respective environments presented in Figure 2, Figure 3 and Figure 4.
> > 	2. For transfer learning, we pretrain on "DoorKey" and on "MultiEnv" respectively. After this pretraining we do transfer learning on all other environments.
> >
> > > "which are diverse in the required skills." (p. 6) ← not sure what this means
> >
> > We clarified this to: "which require the agent to learn a large skill set."
> > > "We see that the cyclophobic intrinsic reward is necessary to reach a successful trajectory at all." (p. 8) Not sure what this means, since the figure only shows a comparison of two versions of cyclophobic agents against epsilon-greedy.  Probably should be rephrased.
> >
> > We clarified this to: "We see that the cyclophobic intrinsic reward is essential
> > to reach a successful trajectory at all since epsilon-greedy using only extrinsic rewards is not able to find a successful trajectory at all."
> >
> > > Typos and grammatical suggestions (no intended impact on score)
> >
> > Thanks a ton for the all the suggestions!  That is very helpful for us.

---

### Author Response · Authors · 2022-11-15
**General comment for all reviewers and area chair**

We are surprised that we received low scores of 3, 3, 5 for our paper, eventhough we reach SOTA results beyond what was possible in NeurIPS 2021 oral-paper Parisi et al.  The reviewers do agree that our ideas are easy to understand, well explained and properly evaluated with ablation studies.

So, what is wrong with our paper?  One reviewer critizes limited novelty, while we do present SOTA results on several difficult tasks.  Another asks whether our results generalize to Atari games, which are out of the scope of Parisi's setup, that we study, as well.

Of course, we are grateful to the reviewers for their time and their useful comments.  We already incorporated them into the revised version and tried to answer the concerns of the reviewers below, which hopefully convinces them of the quality of our paper.

---

> ### Comment · Reviewer_CKV7 · 2022-11-16
> **Never claim SOTA so easily**
>
> Okay, let's talk about SOTA. I would kindly remind the authors that it has been a year since NeurIPS 2021. MiniGrid is a popular testbed, and you may need to make really careful claims when talking about SOTA at this moment. I would suggest the authors check the results in the papers that I have mentioned in my review and even survey some more recent papers before making such a strong claim.
>
> Finally, SOTA isn't the only criterion for a strong paper, either. I agree that the loop closure idea is a good one to pursue, but complaints never make a paper better.
>
> Good luck.

---

> > ### Author Response · Authors · 2022-11-18
> > **Response to "Never claim SOTA so easily"**
> >
> >
> > > I would suggest the authors check the results in the papers that I have mentioned in my review and even survey some more recent papers before making such a strong claim.
> >
> > We have gathered a series of experiments from the other papers that overlap with ours in minigrid (see Sec. A.3). For tabular methods we are definitely SOTA and beat other (not all) neural network based methods. In larger environments it becomes more difficult for our tabular method to compete due to observational complexity (see Sec. 3.3). More details:
> >
> > * **AMIGO**: For ObstructedMaze-1Dl our method converges quicker and more reliably. For KeyCorridor S3R3, while we do not converge with the tabular method the neural network method converges quicker. For ObstructedMaze-2Dlhb we start converging orders of magnitude quicker although performance does not improve after that.
> >
> > * **MADE**: ObstructedMaze2Dlh and Obstructed2Dlhb, our method starts to converge quicker less top end performance. For KeyCorridor S3R3 about the same performance.
> >
> > * **NoveID**: ObstructedMaze2Dlh and Obstructed2Dlhb, our method starts to converge quicker less top end performance. For KeyCorridor S3R3 about the same performance.
> >
> >
> > We think testing on tabular methods is important as it shows that exploration methods must both account for observational complexity and task complexity (see also Sec.3.3).
> >
> > For instance, Parisi et al. show in their experiments that even simple count based exploration works well with neural networks.
> >
> > Furthermore, to support Sec. 3.3,  when we train on only one color in the ObstructedMaze and KeyCorridor environments we are much faster than other methods (see Figure 8). Also in Figure 9 we see that the size of the environment becomes critical for the larges ObstructedMazes.
> >
> > > However, the experiment section is unsound and I'm concerned about the reproducibility of the reported results.
> >
> > It is valid to request MADE, AMIGO and noveID, however given the standpoint of our paper (see answer to other questions) and our method being tabular we disagree with the experiments being unsound. Furthermore, the code will be publicly availabe and we now show error margins for all our results.
> >
> >
> >
> >
> > > Finally, SOTA isn't the only criterion for a strong paper, either.
> >
> > Sure, SOTA is not everything and we are sure there are other good methods out there.  However, we intentionally combined our hierarchical cyclophobic reward with the tabular setup to study its quality independent of deep learning components (see Sec. 3.3).  Of course, focussing on the tabular setup leads to experiments on smaller environments.

---

### Decision · Program_Chairs · 2023-01-20

**Decision:**

Reject

**Justification For Why Not Higher Score:**

Promising results, but evaluation limited to two similar gridworlds, and with missing baselines.

**Justification For Why Not Lower Score:**

N/A

**Metareview: Summary, Strengths And Weaknesses:**

The paper introduces a novel intrinsic reward for exploration based on cycle avoidance, where cycles are detected across multiple agent centric views. Being a tabular method, the method is evaluated on MiniGrid and MiniHack, in both tabula rasa and in a transfer learning regime.

Overall the reviewers found the paper to be well written, the method intuitive and appreciated that the limitations of the method were addressed head-on.

Unfortunately, all reviewers and this AC agree that while promising, the paper is not yet ready for publication. Key points of contention remained throughout the discussion phase. In particular, I agree with [CKV7] regarding the missing baselines: the fact that these papers did not consider transfer learning does not preclude a proper comparison when learning from scratch. At its core, Cyclophobic RL is a standard exploration method which ought to be evaluated in both tabula-rasa and transfer learning settings. I also share [FEf9]’s concern regarding application domains: the authors chose two very similar domains to evaluate their method and it is also not clear to me whether their method would scale to more complex or richer environments, whether Atari, Habitat or other. The choice of agent-centric hierarchical views is central to their method and represents prior knowledge which may or may not transfer or be as easily applicable to other environments. Ideally, the authors should provide convincing evidence their method is broadly applicable.

Given the promising results, I sincerely hope the authors will take the feedback to heart and improve the paper for a future submission.